# Functional regression clustering with multiple functional gene expressions

Susana Conde[1,2,3,4¤], Shahin Tavakoli[5], Daphne Ezer[1,2,3]*

**1** Department of Statistics, University of Warwick, Coventry, United Kingdom, **2** The Alan Turing Institute, London, United Kingdom, **3** Department of Biology, University of York, York, United Kingdom, **4** School of Mathematical Sciences, University of Southampton, Southampton, United Kingdom, **5** Research Institute for Statistics and Information Science, Geneva School of Economics and Management, University of Geneva, Geneva, Switzerland

¤ Current address: Hospital del Mar Medical Research Institute, Barcelona, Spain
* daphne.ezer@york.ac.uk

## Abstract

Gene expression data is often collected in time series experiments, under different experimental conditions. There may be genes that have very different gene expression profiles over time, but that adjust their gene expression patterns in the same way under experimental conditions. Our aim is to develop a method that finds clusters of genes in which the relationship between these temporal gene expression profiles are similar to one another, even if the individual temporal gene expression profiles differ. We propose a *K*-means-type algorithm in which each cluster is defined by a function-on-function regression model, which, inter alia, allows for multiple functional explanatory variables. We validate this novel approach through extensive simulations and then apply it to identify groups of genes whose diurnal expression pattern is perturbed by the season in a similar way. Our clusters are enriched for genes with similar biological functions, including one cluster enriched in both photosynthesis-related functions and polysomal ribosomes, which shows that our method provides useful and novel biological insights.

**Data Availability Statement:** The original data in our manuscript was previously used in: Nagano AJ, Kawagoe T, Sugisaka J, Honjo MN, Iwayama K, Kudoh H. Annual transcriptome dynamics in natural environments reveals plant seasonal

## Introduction

Next-generation sequencing technology (specifically RNA-sequencing or RNA-seq) allows researchers to accurately measure gene expression for all genes in a biological sample [1]. Until recently, it was prohibitively expensive to perform RNA-seq experiments at more than a few time points at once. RNA-seq is now widespread and affordable enough to use it to investigate time-sensitive biological processes, such as response to environmental stimuli or the organism's internal clock [2]. In this context, typical biological questions are to detect genes that are differentially expressed over time, and clustering genes according to their expression time courses. Such clustering efforts have mainly been focussed on finding subgroups of genes sharing common time course patterns [3, 4].

In addition to single RNA-seq time-course experiments, it is now common to perform multiple time series RNA-seq experiments under multiple different treatments [3, 4], and methods

adaptation. Nature Plants. 2019;5:74–83. "The sequence data that support the findings of this study are available in the DDBJ Short Read Archive repository, with the accession numbers DRA005871, DRA005872, DRA005873, DRA005874, DRA005875 and DRA005876, which are all available at https://www.ncbi.nlm.nih.gov/bioproject/PRJDB5830. Database of detailed results of individual genes is at http://sohi.ecology.kyoto-u.ac.jp/AhgRNAseq/".

**Funding:** This project was funded by the Alan Turing Institute Research Fellowship under EPSRC Research grant (TU/A/000017) to DE; Biotechnology and Biological Sciences Research Council (BBSRC) and Engineering and Physical Sciences Research Council (EPSRC). EPSRC/BBSRC Innovation Fellowship (EP/S001360/1) to DE and SC. ST would like to thank the Isaac Newton Institute for Mathematical Sciences, Cambridge, for support and hospitality during the programme Statistical Scalability where work on this paper was undertaken. This work was supported by EPSRC grant no EP/R014604/1. Engineering and Physical Sciences Research Council (EPSRC): https://www.ukri.org/councils/epsrc/ Alan Turing Institute: https://www.turing.ac.uk/ Biotechnology and Biological Sciences Research Council (BBSRC): https://www.ukri.org/councils/bbsrc/ Isaac Newton Institute for Mathematical Sciences: https://www.newton.ac.uk/ The funders did not play any role in the study design, data collection and analysis, decision to publish or preparation of the manuscript.

**Competing interests:** The authors have declared that no competing interests exist.

for analysing such data are not well-established. In this paper, we propose a fundamentally different approach to clustering such multiple gene expression time course, by using the link between them—through a functional regression—as a way of clustering genes. This strategy would be able to group together genes that may have very different temporal expression profiles in different conditions, but whose profiles change in the same way across treatments. We hypothesize that such genes would be part of the same pathways. For instance, two genes might have different gene expression patterns under normal growth conditions, because they are normally regulated by different sets of regulatory proteins. However, both these genes might be regulated by the *same* stress-associated regulatory protein in response to an environmental stimulus, which causes their gene expression pattern to be perturbed in the same way.

Our approach is based on treating gene expression time courses as curves that are sampled discretely, with measurement error, and falls therefore within the realm of functional data analysis (FDA) [5–9], now a prevalent area of statistics, with applications in numerous fields, such as in neuroimaging [10–17], phonetics [18–22], or genomics [23–26]. Classical statistical modelling tools have been extended to functional data: the functional linear model (FLM) has received a lot of attention [27], see the review of and references therein, and many generalizations have been proposed, such as those inspired by generalized linear models [28], (generalized) additive models [29–31], or non-parametric regression [6].

Parallel to this, there have been many contributions to the clustering of functional data: generalizations of $K$-means have been proposed, usually after projecting the functional observations onto some finite basis of functions [23, 32–34], or onto the first functional principal components (FPC) [35]. Extensions of such functional $K$-means have also been proposed: [36] proposed using the subspaces spanned by FPCs as representative of the clusters, instead of using cluster means to define the clusters, and [37] proposed a functional version of the reduced $K$-means algorithm [38]. [39] proposed to choose adaptively the projections onto which the $K$-means algorithm is applied, and showed that the technique can yield asymptotically perfect clustering. Mixture models for functional clustering have also been proposed [40–42].

Combining functional linear models and functional clustering has, however, been far less studied. Such a problem is motivated, for instance, by trying to find subgroups in the data characterized by different relationships between the (scalar or functional) response, and one or several functional covariates. An example of such problem arises in plant genomics, where one is interested in clustering the genes of a plant based on the relationship of its circadian expression in summer, say $Y(t)$, where $t \in [0, 48]$ is a time observation in hours i.e. from a process observed during two entire days (and 0 represents midnight of the first day), and its relationship with the circadian expressions in autumn, winter and spring, say $X_1(t), X_2(t), X_3(t), t \in [0, 48]$. In this setting, one could consider a cluster-specific functional linear model, such as

$$Y_i(t) = \beta_{0k}(t) + \sum_{j=1}^{3} \int_0^{48} \beta_{jk}(t,s)X_{ij}(s)ds + \varepsilon_i(t), \quad t \in [0, 48], \tag{1}$$

if gene $i$ belongs to group $k \in \{1, \ldots, K\}$, where $\mathbb{E}(\varepsilon_i(t)) = 0$ for all $t \in [0, 48]$. The group memberships are of course unknown, but finding clusters based on model (1) is of interest, and gives promising results when applied to gene expressions timecourses of *Arabidopsis halleri* specimen, see later section about gene seasonal data set.

To the best of our knowledge, only a handful of papers have considered clustering functional data using functional models similar to (1). [43] looked at scalar-on-function regression modelling (i.e. the case $Y(t) \equiv Y \in \mathbb{R}$ in (1)) with one functional covariate, and used FPCA for regularization. [44] considered the concurrent functional linear model $Y(t) =$

$\sum_{j=1}^{p} \beta_{jk}(t)X_j(t) + \varepsilon_k(t)$ if the observation comes from group $k$, and used a mixture of Gaussian processes to fit the model using an EM-type algorithm. In this paper, we present a multivariate functional clustering method based on a cluster-specific functional linear model like (1), called Functional Regression Clustering (FRECL). It clusters data of the form $\{(Y_i, X_{i1}, \ldots, X_{ip}): i = 1, \ldots, n\}$, where $Y_i, X_{i1}, \ldots, X_{ip}$ are curves, and allowing $p > 1$. Our proposed method has been developed with the application to gene expression time courses in mind, but it can also be used in other applications, such as the multiple sclerosis applications of [12]. Indeed, FRECL is versatile enough to be easily applied to cluster any data set in which multiple functional data sets are being compared.

The paper is organized as follows. In the next section we describe and present our FRECL model and method for finding the clusters and the regression surfaces $\beta_{jk}(t, s)$. In the next section we provide a description of our motivating application for FRECL. Then we provide an extensive simulation study of the performance of our method in data that resembles that in our motivating example, and compare it to existing (functional) clustering methods. This is followed by a section where we identify clusters in an expression time course data set utilising FRECL, and we conclude with a discussion. The full code of our method is made available for reproducing our results, and for further applications, see https://github.com/stressedplants/FRMM.

## Description of method

Given multivariate functional observations $\{(Y_i, X_{i1}, \ldots, X_{ip}): i = 1, \ldots, m\}$, where $Y_i, X_{i1}, \ldots, X_{ip} \in L^2(\mathcal{T})$, where $L^2(\mathcal{T}) = \{f : \mathcal{T} \to \mathbb{R} : \| f \| < \infty\}$ is the Hilbert space of square integrable functions defined over the compact interval $\mathcal{T} \subset \mathbb{R}$, with norm $\| f \| = (\int_{\mathcal{T}} f^2(t)dt)^{1/2}$, the goal of our paper is to cluster these observations into groups according to the relation between the *responses* $Y_i \in L^2(\mathcal{T})$, and the *predictors* $X_{i1}, \ldots, X_{ip} \in L^2(\mathcal{T})$. While we assume that the functional responses and predictors are all defined on the same domain $\mathcal{T}$, our method can easily be extended to settings with distinct domains for the response and each predictor.

Let $\mathcal{P}_{m,K}$ denote the set of partitions of $\{1, \ldots, m\}$ into $K > 1$ disjoint sets (which we shall call *clusters*), with elements of the form $P = \{C_1, \ldots, C_K\}$. Notice that we allow partitions with empty clusters, which implies that $\mathcal{P}_{m,K-1} \subset \mathcal{P}_{m,K}$ provided we identify sets of the form $\{A_1, \ldots, A_L, \emptyset\}$ with $\{A_1, \ldots, A_L\}$, where $A_1, \ldots, A_L$ are nonempty sets, $L \geq 1$. Let $\mathbb{1}_A$ the indicator function of the set $A$, defined by $\mathbb{1}_A(x) = 1$ if $x \in A$, and $\mathbb{1}_A(x) = 0$ otherwise. We assume that the observations $\{(Y_i, X_{i1}, \ldots, X_{ip})\}$ come from the following model,

$$Y_i(t) = \sum_{k=1}^{K} \mathbb{1}_{C_k^*}(i)\left\{\beta_{0k}(t) + \int_{\mathcal{T}} \beta_{1k}(t, s)X_{i1}(s)ds + \cdots + \int_{\mathcal{T}} \beta_{pk}(t, s)X_{ip}(s)ds\right\} + \varepsilon_i(t), \quad (2)$$

where $P^* = \{C_1^*, \ldots, C_K^*\} \in \mathcal{P}_{m,K}$ is a fixed partition, $\varepsilon_i(t)$ is a functional error term, $\beta_{0k} \in L^2(\mathcal{T})$, and $\beta_{jk} \in L^2(\mathcal{T} \times \mathcal{T})$, $j = 1, \ldots, p$, $k = 1, \ldots, K$. In other words, the same functional linear model links $Y_i$ and $(X_{i1}, \ldots, X_{ip})$ within each cluster $i \in C_k^*$, but the functional parameters are (possibly) distinct across the clusters $C_1^*, \ldots, C_K^*$. The goal is to find the *unknown* partition $P^*$.

Letting $\boldsymbol{\beta}_k = (\beta_{0k}, \beta_{1k}, \ldots, \beta_{pk}) \in L^2(\mathcal{T}) \times L^2(\mathcal{T} \times \mathcal{T}) \times \cdots \times L^2(\mathcal{T} \times \mathcal{T})$ be a functional vector, and defining

$$\Phi((X_{i1}, \ldots, X_{ip}), \boldsymbol{\beta}_k)(\cdot) = \beta_{0k}(\cdot) + \sum_{j=1}^{p} \int_{\mathcal{T}} \beta_{jk}(\cdot, s) X_{ij}(s) ds \in L^2(\mathcal{T}), \tag{3}$$

then (2) can be rewritten as

$$Y_i = \sum_{k=1}^{K} \mathbb{1}_{C_k^*}(i) \Phi((X_{i1}, \ldots, X_{ip}), \boldsymbol{\beta}_k) + \varepsilon_i.$$

Another way of expressing model (2) is to say that a model

$$Y_i = \Phi((X_{i1}, \ldots, X_{ip}), \boldsymbol{\beta}) + \varepsilon \tag{4}$$

holds within each cluster of $P^*$, with possibly distinct functional parameters $\boldsymbol{\beta}$. Note that Eq (4) is just a compact representation of equation (2). $\Phi$ links the covariate $X_{il}$ and the regression slopes $\beta$, so by using a generic $\beta$ in Eq (4) we allow the conditional mean to be equal to any of the conditional means in the previous equations. It is not straightforward to view model (2) as a mixture model since density functions are generally not well defined in a functional context [45]. Notice that (4) is a function on function regression model with multiple functional predictors, and that $\Phi((X_{i1}, \ldots, X_{ip}), \boldsymbol{\beta})$ can be viewed as the conditional expectation of the functional response $Y_i$ given $(X_{i1}, \ldots, X_{ip})$ and $\boldsymbol{\beta}$, provided $\mathbb{E}\,\varepsilon(t) = 0, t \in \mathcal{T}$.

Our strategy for finding $P^*$ is inspired by the $K$-means algorithm [46], which we can view as an iterative procedure alternating between a model fitting (the computation of the cluster means given the cluster allocations) and a partition update (assigning each observation in the next iteration to the current closest cluster mean). We therefore propose to estimate $P^*$ using an iterative procedure, summarized as follows:

1. Pick $P \in \mathcal{P}_{m,K}$ with non-empty clusters at random,

2. Fit model (4) within each cluster of $P$, thus obtaining $K$ fitted models,

3. Reassign the $i$th observation, $i = 1, \ldots, m$ to the best fitting model, which we refer to as $\hat{k}(i)$, a function that returns the assigned cluster designation for each observation:
   $\hat{k}(i) \in \{1, \ldots, K\}$, thus defining a new partition $P^+$,

4. Set $P \leftarrow P^+$ and repeat steps 2–4 until convergence.

Notice that step 2 requires a fitting method $\mathcal{F}$. We discuss the choice of $\mathcal{F}$ below. Step 2 results in $K$ fitted models of the form (4), with estimates $\hat{\boldsymbol{\beta}}_k$. Step 3 requires finding the best fitting model for each observation $(Y_i, X_{i1}, \ldots, X_{ip})$; we choose to do this by computing the norm of its fitted residuals under each model,

$$\hat{r}_{ik} = \| Y_i - \Phi((X_{i1}, \ldots, X_{ip}), \hat{\boldsymbol{\beta}}_k) \|, \quad k = 1, \ldots, K, \tag{5}$$

Other methods for updating clusters can be chosen, such as choosing a different norm in (5). The full version of our generic FRECL algorithm, using the fitting method $\mathcal{F}$, is given in Algorithm 1.

**Algorithm 1:** Generic FRECL algorithm run

**Input:** $K > 1$, data $\{(Y_i, X_{i1}, \ldots, X_{ip})\}$, and method $\mathcal{F}$ for fitting model (4), Stopping criterion $\mathcal{S}$.
**Result:** A partition $P \in \mathcal{P}_{m,K}$.

```
begin
  Pick at random an initial partition P₀ = {C_{0,1},...,C_{0,K}} ∈ 𝒫_{m,K} with non-
  empty clusters,
  j ← 0.
  repet
    Fit model (2) for partition P_j using method ℱ:
    for k = 1, ..., K do
      Compute the estimates β̂_k from the data {(Y_i, X_{i1}, ..., X_{ip}):i ∈ C_{j,k}}
      using fitting method ℱ,
      C_{j+1,k} ← ∅.
    end
      Reallocate each observation to the best fitting model:
      for i = 1, ..., m do
        for k = 1, ..., K do
          Compute r̂_{ik} as in (5).
        end
        Compute k̂ ← argmin_{k=1,...,K} r̂_{ik},
        C_{j+1,k̂} ← C_{j+1,k̂} ∪ {i}.
      end
      P_{j+1} ← {C_{j+1,1}, ..., C_{j+1,K}},
      K ← "Number of non-empty partitions in P_{j+1}",
      j ← j + 1.
  until 𝒮 is true.
  Return final partition P_j ∈ 𝒫_{m,K}.
end
```

Because the output of Algorithm 1 depends on the initial random partition, we propose to use consensus clustering [47] to produce more consistent results. Consensus clustering consists of running a clustering algorithm multiple times, with different initial partitions, and then aggregating the obtained clusters. We describe it formally in Algorithm 2.

**Algorithm 2:** Complete FRECL algorithm, with consensus clustering.

```
Input: K > 1, L > 1, and the input for Algorithm 1.
Result: A partition P ∈ 𝒫_{m,K}.
begin
  for l = 1, ..., L do
    Run Algorithm 1. If convergent, denote its resulting partition P_l;
    otherwise discard the run. Let A^{(1)} be the m × m binary matrix with
    (i, j)th entry equal to 1 if i, j are clustered together (according
    to P_l), and zero otherwise.
  end
  Compute the consensus matrix B = ∑_{l=1}^{L} A^{(1)},
  Perform K-means clustering with the rows of B as observations, and
  return the resulting partition.
end
```

Algorithms 1 and 2 depend on a couple of parameters, which we now briefly discuss and make recommendations about. A more detailed discussion of the choice of some of these parameters is deferred to the Simulation Section below.

## Fitting method $\mathcal{F}$

Fitting functional linear models (FLM) involves solving ill-posed inverse problems, and requires some form of regularization, which is generally performed by projection of the functional observations on a finite number of functional principal components. This is usually performed either after transforming the discretely observed functional data into curves, or simultaneously, see [27, 48] for overviews of functional regression. A more recent approach to

FLM is to use computational methods—such as boosting [49]—for model fitting. [50] propose such approach, which is nicely implemented in the R package FDboost [51, 52]. In the rest of the paper, we use $\mathcal{F}$ as given by the R function FDboost, which regularises automatically the functional regression fit. We will always work with the discretized.

## Stopping criterion $\mathcal{S}$

Currently, our stopping criteria is when (i) either convergence is reached, i.e. the partitions are the same between two consecutive iterations or alternatively (ii) the number of iterations has exceeded a fixed value which we set to 300. Several alternative approaches are possible. For instance, a stopping criteria may be very strict, such as "only stop iterations when either convergence or a cycle is reached." It may also be possible to stop at some point which is approaching convergence, when only a small proportion of observations change clusters across an iteration. Our choice of stopping criteria is motivated by the analysis in Section about convergence properties.

## Number of clusters $K$

In order to determine the number of clusters, we propose to run the algorithm for a range of values of $K$, and compute each time the mean squared error (MSE) with the residuals from the final partition $P = \{C_1, \ldots, C_K\}$:

$$\mathrm{MSE}(K) = \frac{1}{m} \sum_{k=1}^{K} \sum_{i \in C_k} \hat{r}_{ik}^2 \tag{6}$$

where $\hat{r}_{ik}$ is as in (5). We then plot those quantities and use an elbow-like criterion.

## Number of runs $L$ for consensus clustering

The choice of $L$ depends on the balance between runtime and robustness. The larger the value of $L$ the longer it takes for the algorithm to complete, but the more consistent the clusters will be. We analyse the impact of varying this parameter in more depth in Section about the power of consensus clustering.

## Motivating application

### Biological background

FRECL can cluster any data set in which each observation consists of two or more functional data. However, we were specifically interested in developing this method to provide new biological insights related to how plant gene expression changes over time in response to the seasons. In our analysis, we aimed to find clusters of genes determined by associations between daily gene expression patterns during the summer, and those during autumn, winter and spring. Many agriculturally-relevant traits that are of interest to plant biologists, such as flowering, occur in the summer. However, many of the developmental decisions that lead to these traits are thought to occur in the other seasons. For instance, flowering time in the summer is determined by the temperature in winter (vernalisation) [53] and the changes in day length in the spring (photoperiod sensing) [54, 55].

About a third of genes are controlled by the circadian clock and vary their expression levels over the course of the day [56]. Indeed, many of the key vernalisation and photoperiod sensing genes are known to be directly regulated by the circadian clock [57]. In many species, the daily pattern of gene expression varies across different seasons because of differences in day length,

vernalization and winter-dormancy [54, 55, 58]. Even genes that are not regulated by the circadian clock may be more sensitive to environmental fluctuations, like pests, shade or UV light, during specific seasons [59]. Also, some genes may play one biological role in one season and play another role in another season, especially genes involved in plant development and response to plant hormones.

Because of these properties, we thought that genes that are biologically associated with one another may have very different expression patterns, but may have similar gene expression changes across different seasons. If this were the case, we would expect FRECL to produce clusters of genes that share biological roles.

### Description of data

The publicly available gene expression data [60] was collected to investigate how diurnal patterns of gene expression change in different seasons. It contains gene expressions of 32669 genes from an experiment done in *Arabidopsis halleri* specimens, a perennial relative of the model plant organism *Arabidopsis thaliana*. The expressions were measured via RNA-seq at four seasons (winter/summer solstice and spring/autumn equinox), over the course of 48 hours, sampled every other hour, with 5–6 replicates per time point. See additional details in Appendix.

### Pre-processing data set

The following pre-processing steps were undertaken before this data was used in either Sections about Simulations or about application to new data.

We computed the median gene expression value over the replicates per time point for each season. Some genes were lowly expressed in nearly all time points. To filter these out, we selected genes that were expressed at moderate levels in 20 or more time points in each season. We define moderate expression levels as those that surpass 5 transcripts per million (TPM), which is a unit of gene expression after normalising RNA-seq data by the sequencing library depth and gene size. Using a TPM threshold of 5 is a common strategy used in biology for filtering out very lowly expressed genes [59].

We transformed all the variables by subtracting, for each gene, the sample mean curve of all the gene expression, and then smoothed the resulting curve by using locally estimated scatterplot smoothing (LOESS) [61, 62], formed with local quadratic polynomials. For the $i$th gene, let $Y_i(t)$ represents the (median, transformed, LOESS-ed) gene expression at time $t$ in summer, and let $X_{ij}(t)$ be its expressions in spring, autumn and winter for $j = 1, 2, 3$, respectively. We drop the words "transformed, median, LOESS-ed" from now on. For computations, we used the evaluation each smoothed curve at the original set of time points.

We describe a simulation study that was designed so that the data had very similar properties to the motivating example. We apply FRECL to the real gene expression data set, and demonstrate that this method is useful for generating new biological insights and hypotheses for future investigations.

## Simulations

### Simulation strategy

**Overview.** We compare our method with these alternative approaches in a simulation study in the Subsection about comparison with other methods. The simulated data was generated to represent realistic situations, so that the results would be applicable to real data. In order to generate a new simulated data set that shares many of the properties of the real one,

we sample the explanatory variables and assign them to known partitions. Additionally, we use the explanatory and response variables to generate a set of realistic model parameters, which we sample from when we assign parameters to each partition. Finally, for each partition, the sampled explanatory variables and parameters are used to generate simulated response variables.

**Generating the simulated data set.** We want to simulate data generated from the model in Eq (2), using the real data $D = \{(Y_i, X_{i1}, X_{i2}, X_{i3}), i = 1, \ldots, m\}$ described in the previous section. The model has then a functional response, which is the gene expression at summer, and $p = 3$ explanatory variables, which are the gene expressions at spring, autumn and winter. We work with the discretized variables defined in the original set of time points $T_0 = \{t_1, \ldots, t_T\} \subset \mathcal{T} = [0, T], T = 24$. The time points, equidistant, are $t_i = i$. First, we choose $\boldsymbol{\beta}_k, k = 1, \ldots, K$ as the fitted parameters obtained by applying full FRECL (and using FDboost [49, 51] for fitting) to the real data $D$, with $K$ equal to the number of desired clusters. Consequently, our choice of parameters represents a realistic situation. We then draw a partition $P^* = (C_1^*, \ldots, C_K^*) \in \mathcal{P}_{m,K}$ at random, which will represent the true clusters. The FDboost package gives the discretized conditional expectation $\Phi$ via the 'predict' routine—i.e., evaluated in $T_0$. Finally, we construct the vector of the discretized simulated response $Y$, by adding errors varying across simulations.

**Scenarios evaluated.** First, we consider the scenario composed by $K = 3$ clusters, independent and identically distributed standard normal random error terms $\varepsilon_i(t) \sim N(0, 1)$, and generate 50 simulations for sample sizes $n = 500, 1000$. Additionally, we perform a sensitivity analysis for a variety of different scenarios. These include varying:

(i) the distribution of the random error term from a discrete version of model (2). We consider two scenarios, both with $K = 3$ and for $n = 500, 1000$. The first one with errors following an auto regressive model with 1 lag (AR1), i.e.

$$\varepsilon_i(t_q) = \rho \, \varepsilon_i(t_{q-1}) + \epsilon_i(t_q), \tag{7}$$

where $\rho = 0.5$ and the innovations are $\epsilon_i(t_q) \sim N(0, 0.1)$, $q = 2, \ldots, T$, $i = 1, \ldots, m$. The second one, with independent and identically distributed standard normal errors, i.e. $\varepsilon_i(t_q) \sim N(0, 1)$, $q = 1, \ldots, T$, $i = 1, \ldots, m$.

(ii) the number of clusters $K = 3, 6, 9, 12$, with an analogous AR(1) random error term and for $n = 500, 1000$;

(iii) the sample size $n = 500, 1000$;

(iv) whether the $L_1$ or $L_2$ norms are used in FRECL, see Eq (5), to quantify the magnitude of the fitted residuals in an iteration. Recall that for a function $f(s)$, $s \in \mathcal{T}$, its $L_1$ norm is given by $\int_{\mathcal{T}} |f(s)| ds$ whereas its $L_2$ norm is $\left[ \int_{\mathcal{T}} (f(s))^2 ds \right]^{1/2}$. We generate 50 simulations for each of the scenarios with $K = 3$; these results are included S2 Fig. As the $L_1$ and $L_2$ norms performed nearly identically, we chose to only use the $L_2$ norm in the remainder of the manuscript.

(v) the number of iterations in the runs of one instance in Algorithm 1, for $K = 3, 12$, and $n = 500, 1000$;

(vi) the number of runs, $K = 12$, $n = 1000$, AR(1) random error term with $\rho = 0.5$, $\epsilon \sim N(0, 0.1)$.

Fig 1 (i, iii), Fig 2 (i, iii), Fig 3 (i, iii) and Fig 4 (ii, iii); are developed in the sections about simulations with 50 replicates and the results from a sensitivity study, Fig 5; (v) is developed in

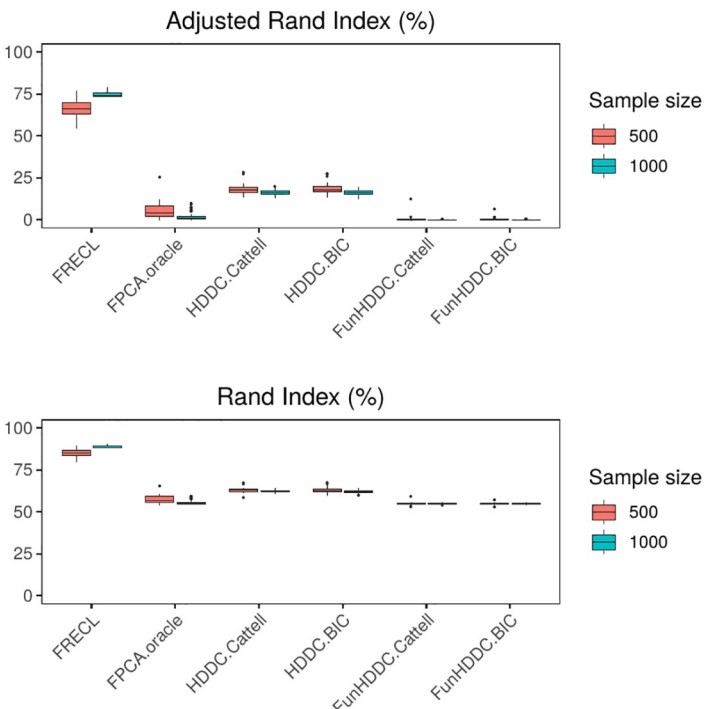

**Fig 1. Distribution of the adjusted Rand index, above, and Rand index, below, for the 50 simulations from a model with $K = 3$ clusters, i.i.d. random error term.**

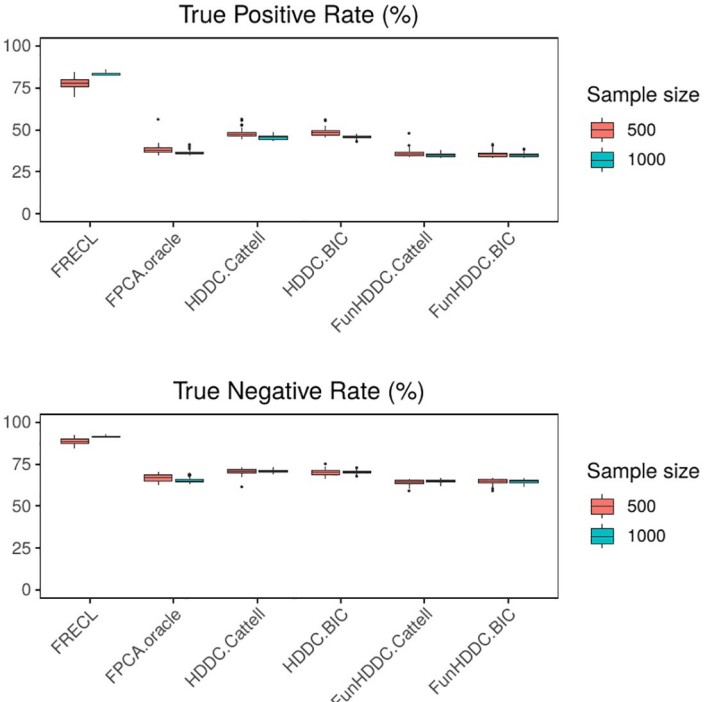

**Fig 2. Distribution of the True Positive Rate, above, and True Negative Rate, below, for the 50 simulations from a model with $K = 3$ clusters, i.i.d. random error term.**

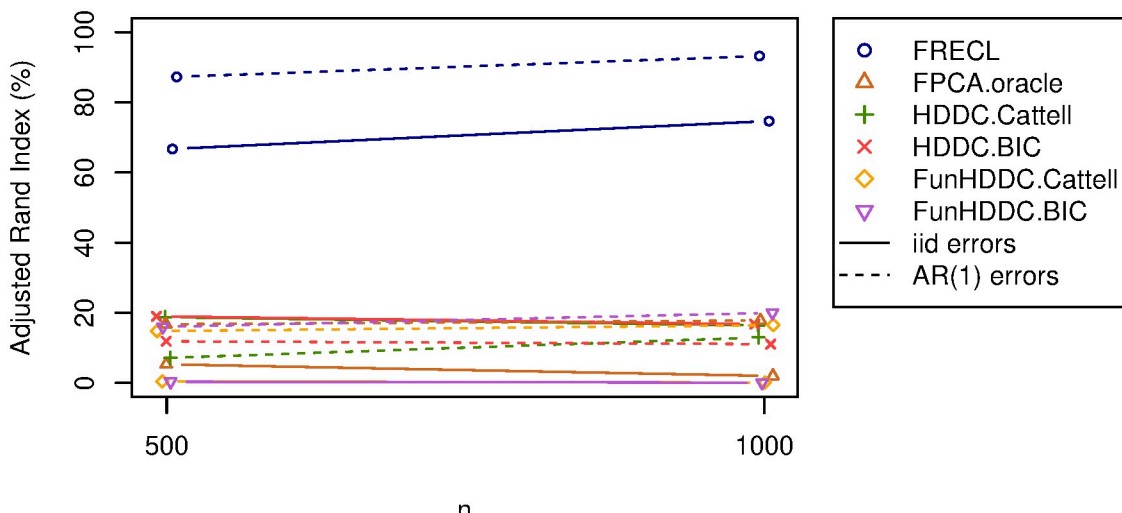

**Fig 3. AR(1) error term, $\rho = 0.5$, $\sigma^2 = 0.1$ versus i.i.d., $\sigma^2 = 1$ for all the methods; $K = 3$.** Lines corresponding to the same algorithm have the same colour. Different error terms distributions are distinguished by line types. A continuous line represents i.i.d. error, an a dotted one, AR(1).

a section about convergence properties of FRECL Fig 6; (vi) is developed in the section about the power of consensus clustering.

**Metrics for evaluating accuracy.** For each simulated data set, we compute the observed adjusted Rand Index (ARI) [63], which allows for chance assuming that the underlying random variable defining the counts of pairs of observations belonging or not to clusters of the two partitions that are being compared is hypergeometric, the Rand index [64], which is the proportion of correctly classified pairs of observations selected at random, the true positive and the true negative clustering rates in the space of all pairs of observations. Specifically, we

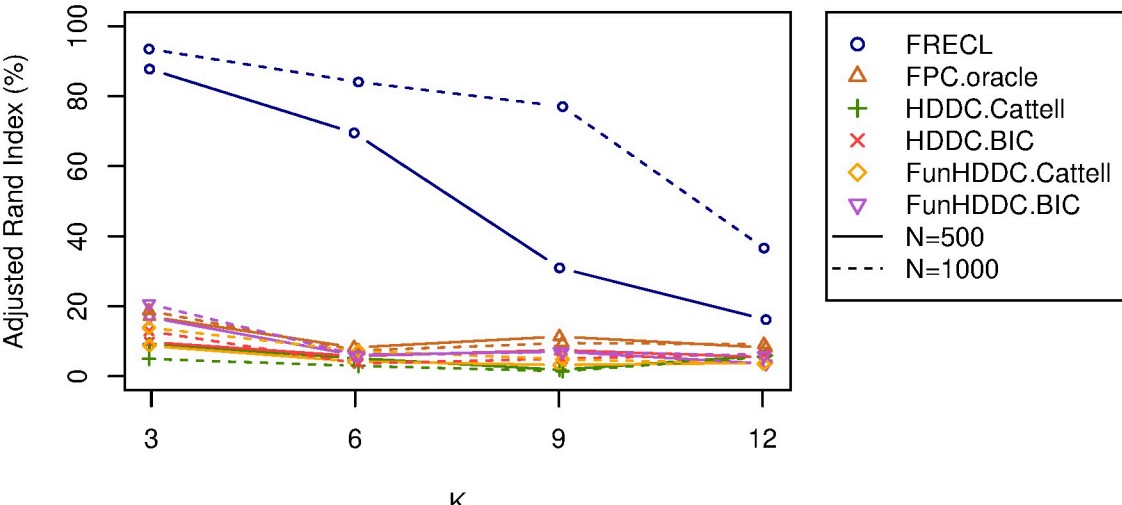

**Fig 4. Observed adjusted Rand index for one-replicate simulations (with AR(1), $\rho = 0.5$, $\sigma^2 = 0.1$) against the number of clusters $K = 3, 6, 9, 12$ (on the horizontal axis).** Lines corresponding to the same algorithm have the same colour. Sample sizes are distinguished by line types.

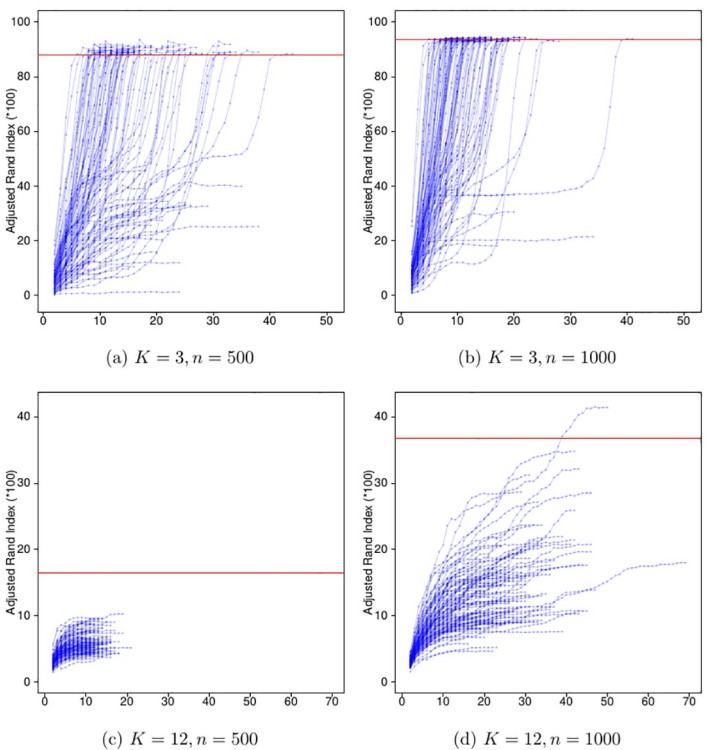

(a) $K = 3, n = 500$ (b) $K = 3, n = 1000$

(c) $K = 12, n = 500$ (d) $K = 12, n = 1000$

**Fig 5. Observed adjusted Rand index for iteration in FRECL, 100 runs.** Each line represents the values for a run. The ARI is usually monotonic increasing but not completely. Small underlying number of clusters sizes results in better performance. Moreover, the ARI increases very steeply in all the iterations close to the last one, which suggests not to stop FRECL before reaching convergence for speeding up the time. The situation for bigger number of clusters is the opposite. The red lines indicate the value of the final ARI for FRECL after performing consensus clustering on the individual runs.

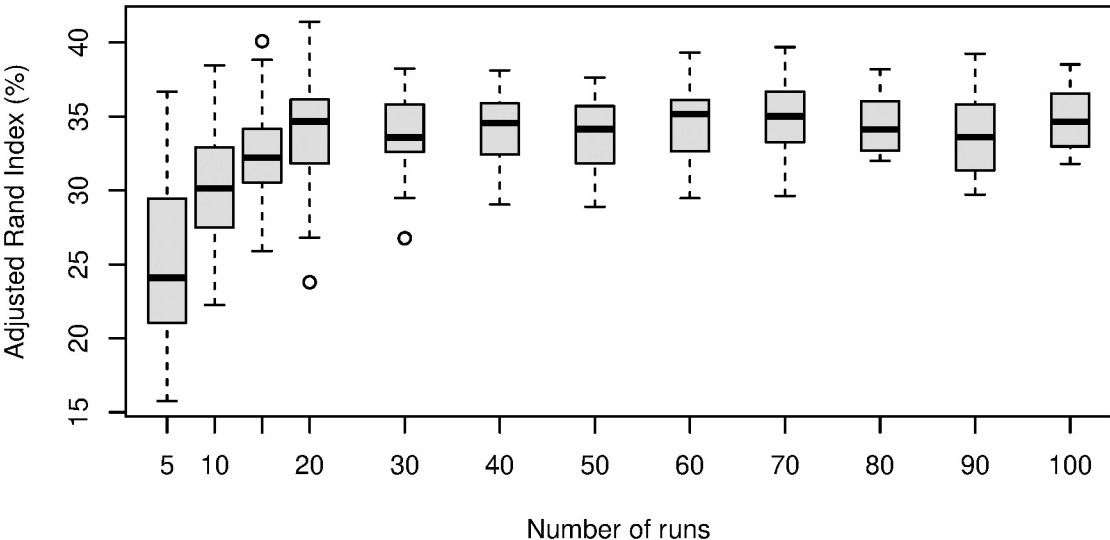

**Fig 6. Observed distributions of ARI ×100 from the FRECL partition after consensus clustering; against the number of runs.** Simulation with $K = 12$ clusters, $n = 1000$, AR(1) random error term.

define a true positive and a true negative as the successful identification of a pair that are or are not part of the same cluster respectively.

**Comparison with other methods.**   Whilst some FDA model developments involve multivariate functional data including at least a functional response and a functional explanatory variable [65], FRECL is the only algorithm we are aware of that clusters observations based on the association between functional explanatory variables and functional response variables. However, there are a number of other methods for clustering functional data, which cluster the observations based on their (functional) response variables $Y$, i.e. ignoring $X = (X_1, \ldots, X_p)$.

We compare FRECL with five such methods. Because of our strategy for generating the simulated data, we would not expect $X$ to be involved in forming any discernible clusters (as these observations are randomly sampled), unlike $Y$ (see the Simulation section for more details). When we compare methods for the simulated data, we use $Y$ only in the main text, but the methods were also evaluated with $X$ and $Y$ appended together, which is shown in S2 Table. When comparing these methods on the real data, we consider observations formed with the values of both the explanatory and response, since $X$ might contain information pertinent for clustering and we did not wish to give FRECL an unfair advantage.

Firstly we compute the functional principal components [5], using a basis of 12 B-spline functions of order 4 with equally spaced knots. The coefficients of the first $s$ principal components are clustered with the $K$-means algorithm, $s = 2, \ldots, 12$ and we select the $s$ that maximises the adjusted Rand Index [63]. This optimal value of $s$ is, of course, unknown, but it can be used as an oracle. We call this method FPCA.oracle.

Secondly, High Dimensional Discriminant Clustering (HDDC), is based on [66] and described as filtering in [67] because the functional observations are approximated by a finite basis of functions. These assume a set of "multivariate" variables, formed in our context by the discrete set of time points. It is a model-based clustering focussed on a Gaussian Mixture model and using the expectation-maximisation algorithm for inference. We consider the default option, which implies selecting the dimension of the FPC space for each cluster using Cattell's test. We call this clustering method HDDC.Cattell. We also consider an alternative to this method, by selecting the FPC space dimensions using the Bayesian Information Criterion (BIC) [68], and call this method HDDC.BIC.

Finally we consider FunHDDC [69, 70] which is a generalisation of HDDC for functional data. It is adaptive because the coefficients of the bases of functions are assumed random variables having a cluster-specific probability distribution [67]. It assumes an underlying latent functional mixture Gaussian model, where, unlike ours, the response is the only (functional) variable, and it models coefficients of basis expansions chosen to be the functional principal components from a cluster-specific analysis. These scores are assumed Gaussian with certain parameters. After estimating the dimensions of the FPC spaces, the model is fitted with the EM algorithm. This method represents an extension of [71]. When these dimensions are estimated with Cattell's test, we call the method FunHDDC.Cattell. Otherwise, when considering the BIC, we call the method FunHDDC.BIC. We initialise the EM algorithm in the FunHDDC approaches with $K$-means.

## Simulation results

**Simulations with 50 replicates and $K = 3$.**   Figs 1 and 2 display boxplots with the distributions of the performance measures for the simulations with $K = 3$ clusters, $L_2$ distance, i.i.d. random error model term, $n = 500, 1000$. In all the cases, FRECL outperforms any other method. It is the only method for which the average performance increases as $n$ increases.

We also note that the FunHDDC approaches, which represent developments specific for clustering functional data, find the poorest results. This outcome is reversed in most of the scenarios of the AR(1) simulated models, see S1 Table in Supplementary File. A one-tailed $t$-test indicates that the ARI in FRECL is a significant improvement over the alternative methods for any sample size ($p < 0.0001$ for any $n$).

**Results from sensitivity study.** Fig 3 compares simulations with a lag 1 autoregressive model to those from an independent and identically distributed error term; for FRECL and the five alternative methods. We note that the two simulated data sets here, for each of the sample sizes, were generated from functional models with the same $\boldsymbol{\beta}$. First, we see that FRECL, which is developed assuming a functional linear model, outperforms any other approach for any of the two simulations and sample sizes. FRECL, FPCA.oracle, FunHDDC. Cattell, and FunHDDC.BIC found greater mean ARI for the AR(1) models regardless of the sample size. Unlike HDDC.Cattell and HDDC.BIC, where the models with an i.i.d. error term have greater mean ARI for $n = 500$ compared to $n = 1000$. For small sample size, FPCA.oracle and HDDC.BIC maximise the mean ARI in AR(1) and i.i.d. models (16.72% and 18.98% respectively). For big sample sizes, FunHDDC.BIC and HDDC.BIC outperform other alternative methods in AR(1) and i.i.d. models (19.98% and 16.71% respectively). FRECL performs outstandingly in all scenarios.

Fig 4 displays the observed ARI against simulations with $K = 3, 6, 9, 12$ clusters, on the horizontal axis, $n = 500, 1000$, for FRECL and the 5 alternative methods. Continuous and dotted lines indicate values for simulations with $n = 500, 1000$ respectively. Overall, all the clustering performances decrease as the number of underlying clusters increases. FRECL outperforms any other approach in all instances. We tried the FunHDDC methods a number of times when these did not find a convergent model, in order to initialise differently the EM algorithm. If it was not possible to find a convergent model with $K$ clusters, we explored models with less number of clusters. Thus, FunHDDC.Cattell found a convergent model with 4 clusters for $K = 6$ with either $n$, with 2, 4 clusters for $K = 9$, $n = 500, 1000$ respectively, and likewise for $K = 12$. FunHDDC.BIC found a model with 7 clusters for $K = 9$, $n = 500$, with 5 clusters for $K = 12$, $n = 500$ and with 11 clusters for $K = 12$, $n = 1000$.

**Convergence properties of FRECL.** The first steps in FRECL consist of performing an iterative procedure a certain number of times prior to computing the consensus matrix. It is of interest to study the evolution of our performance measure of choice (e.g. the adjusted Rand index) for each iteration in a run. If we know that, in a particular scenario where FRECL computation has a lot of burden, as the number of iterations increases, it appears a plateau for our performance measure, then we can speed up the computations by shortening their number.

We computed the adjusted Rand index by iteration for a simulated data set in each of the scenarios: $K = 3, 12$ clusters, $L_2$ distance, $n = 500, 1000$, AR(1) random error terms. Fig 5 depicts the ARI in percentage form. Overall the ARI increases by iteration in all runs. We observed that it is monotonic increasing up to an iteration very close to the last one, e.g. the antepenultimate. When the number of clusters increases, the algorithm has less accurate performance. However, in this scenario the line approximately reaches a plateau when close to the iteration where convergence is achieved. In contrast, for small number of clusters, the values of the ARI increase very steeply towards the 'end' of the lines, suggesting that it is not advisable to shorten the number of iterations in these cases. As $n$ increases, the performance measures increases on average in either scenario.

**The power of consensus clustering.** When searching partitions with larger numbers of clusters, FRECL becomes computationally intensive. It is therefore of interest to investigate whether considering a small number of runs we can achieve an acceptable solution. For this purpose, we set up a study of the distributions of the adjusted Rand Index with different

numbers of runs in one of our simulated data sets corresponding to a scenario with $K = 12$, $n = 1000$, $L_2$ distance, AR(1). We computed the ARI for 20 groups of various numbers of runs, such that all the runs in these groups did not coincide with each other. And all this for 10, 20, 30, . . ., 100 runs as well as for 5 and 15 runs. Fig 6 includes boxplots of these distributions. The variability of the ARI decreases overall as the number of runs of the first stage in FRECL increases. Furthermore, on average it increases, and it becomes convergent from 20 or 30 runs on. This finding indicates that, first, consensus clustering is working because with bigger numbers of runs we obtain better performance, and moreover, we do not need to consider a big number of runs in order to accurately find a partition with a big number of clusters. A 0.05 level t-test gives poor evidence of any difference in the means (with a statistic of −1.2, 95% confidence interval of [−2.53, 0.63], p-value of 0.2323).

## Novel biological insight with FRECL

### Gene seasonal data set

Our central aim was to identify sets of genes whose circadian gene expression profiles in the summer (during the flowering phase) was linked in the same way to gene expression in the autumn, winter, and spring. We expected that there would be modules of genes that may have very different expression patterns from one another, but whose gene expression patterns change in the same way as the seasons progress.

FRECL was used to generate clusters in the seasonal *A. halleri* gene expression data sets previously described and the number of clusters $K$ was determined by the elbow method (see Fig 7(A)). We performed FRECL clustering twice: once using the gene expression profiles over two consecutive days, and once taking the average expression curve across the two days. In each setting, this produced 10 relatively evenly sized clusters (Fig 7(B)).

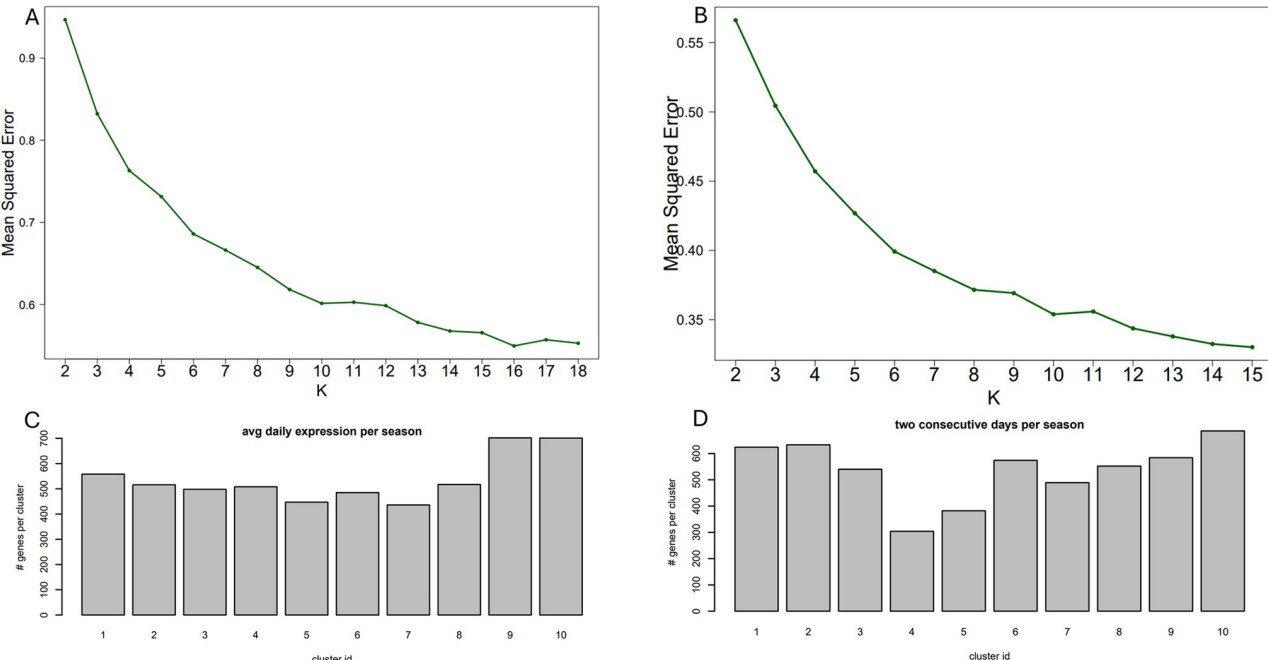

**Fig 7.** On the basis of the elbow method, we selected K = 10 clusters, both for the data set with 23 time points spread over one day (A); and for the data set with 24 time points spread over two days (B). The clusters are approximately evenly sized, under both conditions tested: over one day (C) or two days (D).

Genes in similar pathways do seem to group together in FRECL, suggesting that our method is a useful tool. In many model organisms, each gene is associated with a series of labels representing the cellular components, molecular functions, and biological processes that the protein encoded by the gene is involved in; these are referred to as gene ontology (GO) terms. For every A. halleri gene clustered by FRECL, we identified the most similar gene in the model plant *Arabidopsis thaliana* and searched for GO terms that were significantly associated with each cluster using gProfiler [72]. We successfully identified a number of GO terms that were specific to certain gene clusters, see supplementary spreadsheet for a summary, which indicates that FRECL produces biologically interesting clusters. The adjusted p-values of a few key GO terms are illustrated in Fig 8. Interestingly, we find that ribosome and photosynthesis related genes tend to cluster together in FRECL, which may suggest that the same set of genes regulates the season-dependent gene expression of both processes, suggesting an avenue of research for biologists. Intriguingly, a recent manuscript highlights that both ribosomal processes and photosynthesis are downregulated during plant early age-related senescence, the process by which plants plan leaf death to enable energy expenditure in reproduction [73]. As the seasons progress, we would expect that the plants will mature, so our results are consistent with these findings. In addition, this highlights a strength of our approach: normally ribosome and photosynthesis genes would not be clustered together due to having distinct temporal expression patterns, but we correctly identify that these two processes both change in the same way over the larger temporal scale. We also observe cluster-specific enrichment in polysome, mRNA processing, and immunity-related processes.

The clusters produced by FRECL are very different from those produced by other methods, based on the ARI between the clusters produced by different clustering methods (S5 Table), indicating that our method potentially provides new biological insight. In fact, similarity in clustering algorithms was quite low (S5 and S7 Tables in the supplementary file), suggesting that each of these methods produces very distinct clusters in the real data, perhaps indicating that the partition to clusters in this data is highly dependent on how clusters are defined. A unique aspect of FRECL in comparison to other clustering algorithms is that FRECL clusters on the basis on the relationship between curves, and not only their shape. Indeed, we observe that genes that are assigned to the same cluster do not have similar gene expression patterns with each other, despite sharing GO terms (Fig 8).

Additionally, we were interested in determining whether FRECL provides biological information that can can help us identify gene pairs that share biological roles that would not have been identified using existing clustering methods. There were 184,605 pairs of genes that were found to be in the same clusters in FRECL, but not in any of the other methods (or 301,596 pairs when the average curve was used for clustering). Of these pairs, 169,605 were between pairs of genes whose orthologs in *A. thaliana* were included in AraNet, a gene network in Arabidopsis that uses a Bayesian approach to combine -omics data sets from various organisms to predict functional associations between pairs of genes in Arabidopsis [74, 75] (or 278207 pairs when the average curve was used for clustering). 1470 of the novel pairwise associations found using FRECL but not in any of the other alternative clustering methods were found to be associated with one another in AraNet (or 2330 pairs when the average curve was used).

## Discussion

Whilst [69] claim that FunHDDC, a 'specific' method for clustering multivariate functional data, also works for univariate functional data, our simulation results show that it is outperformed by the other approaches we consider; even by those developed for non-functional,

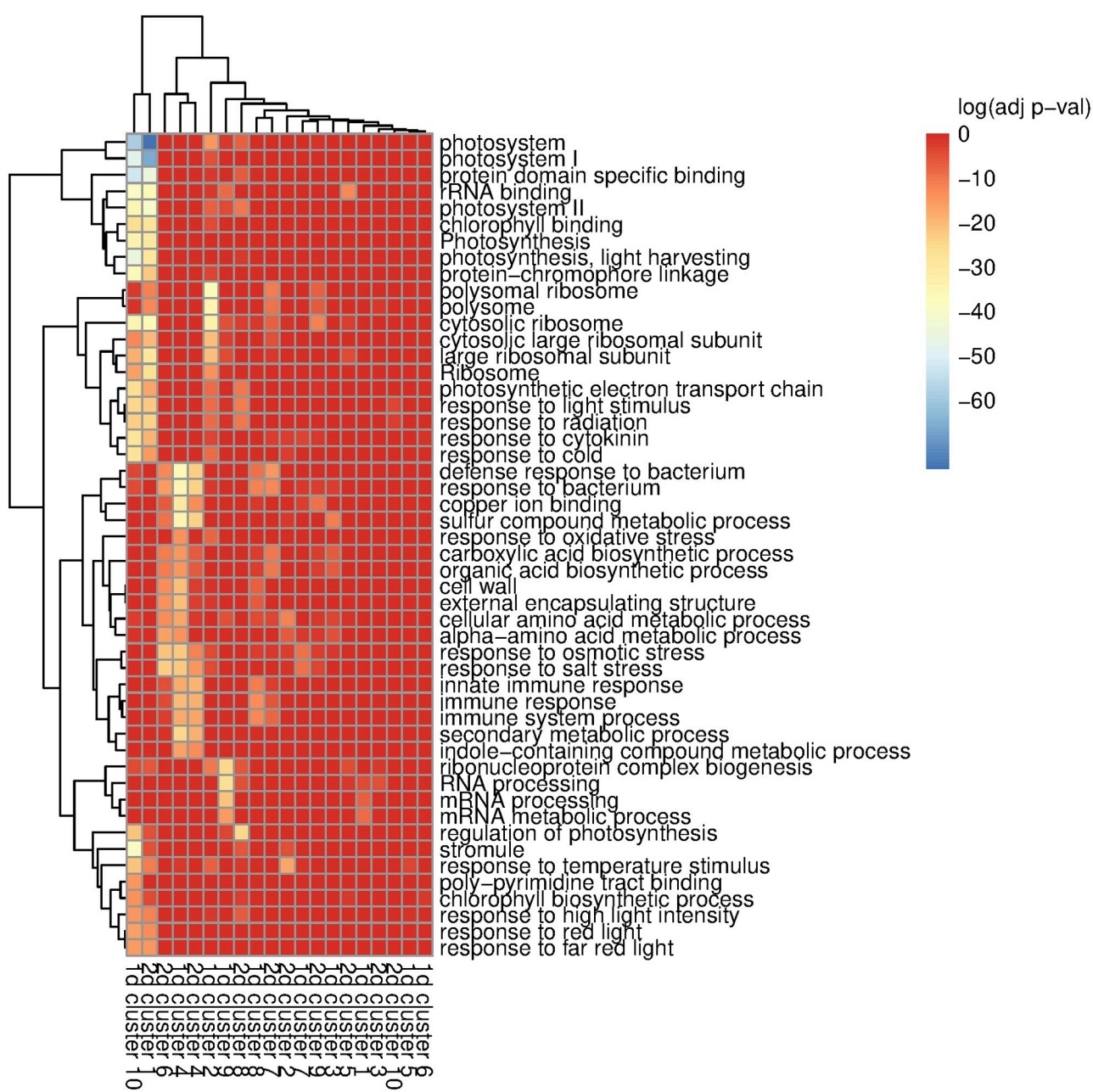

**Fig 8. Each column represents a cluster designated by FRECL, either when considering the average across the two days (1d) or the two days separately (2d).** A heatmap of a selection of biologically interesting GO terms are shown, along with their adjusted p-values based on a gProfiler analysis [72]

high dimensional variables such as HDDC. In one of the illustrations included in [70], FunHDDC is outperformed by HDDC, too.

Functional mixture models make it possible to study relationships between explanatory and response variables over time allowing for clusters characterized by these relationships. [76] presents a clustering method for mixture regression that involves a penalised likelihood, where the penalty is the total entropy. FRECL is an alternative to these settings that does not need to consider a constrained optimization problem, and consequently does not need to estimate the

value of the Lagrange multiplier. [43] propose a functional mixture model implemented with FPCs in order to overcome overfitting with a finite number of observations and infinite-dimensional parameters and as usual selecting a certain number of components. The FPCs are necessarily computed (only) with the explanatory variables. Since the estimation of the slope parameters involves only the same number of selected FPCs, as well, the slope parameter space is restricted, and thus their estimates may be far from the truth. FRECL, clustering based on the functional 'mixture' model (2), improves existing methods, although these were not developed specifically for generating processes with an underlying functional mixture model. Consequently, it is a step further in the development of functional data clustering.

There are several avenues by which FRECL could be extended in the future. For instance, our current implementation assumes that the response and predictors have the same domain. However, it will be possible to extend the method by first scaling the time domains, an extension that would not fundamentally alter the algorithm. Selection of hyperparameters, such as the choice of using $L_1$ and $L_2$ norms may depend on the specific application and will need to be re-assessed when applying the method to new data sets. However, our simulation results suggest the following guidelines: (i) A choice of $L_1$ or $L_2$ norm does not appear to have a large impact on the outcome. (ii) If possible, proceed with each run until convergence. (iii) Increase the number of runs until the consensus clustering converges. (iv) Select the number of clusters using a classical method, like the edge method.

A downside of FRECL is that it is computationally intensive, specifically Algorithm 1. The runtime of Algorithm 1 of FRECL is a product of the number of runs, the number of iterations per run, $K$, and the runtime of the selected functional regression algorithm (which inherently depends on the size of the dataset). The number of iterations per run until convergence is not easily empirically calculated, although it can be experimentally determined for a specific data set, as we have shown. The number of iterations per run until convergence depends on $K$ and the size of the dataset. However, parallelisation is easily implemented, as each run can be computed on an independent node on a computing cluster.

## Software

Software in the form of R code, together with a sample input data set and complete documentation is available at https://github.com/stressedplants/FRMM.

## Appendix

### Data

The seasonal data set contains 3 replicates per time points in autumn and 4 replicates for all the other time points for 32745 genetic entities, 32669 of which are genes. The expression for the $i$th gene is the observed proportion of messenger RNAs of the $i$th gene from specimen X over the total of mRNAs in specimen X multiplied by $10^6$. We removed one suspicious replicate in a time point; it has zero values in many of the genes. If included, the line plots of many of the raw spring gene expressions have a very odd minimum at a time point. We computed the median gene expression per time point as, if using the mean, there are a lot of "ups and downs" in nearby time points. We selected genes whose median expression was >5 units except at most 5 time points in each of the 4 seasons. The first time points are collected at 16:00 hours in spring (March, days 19–21), summer (June, days 26–28), autumn (September, 24–26) and winter (December, 24–26). With these criteria, the resulting data set has $n = 5378$ genes.

## Software

HDDC.Cattell, HDDC.BIC, FunHDDC.Cattell, FunHDDC.BIC are implemented in the `HDclassif` [66] and `FunHDDC` [77] R packages. Curve smoothing was performed using the `fda` [78] R package.

## Supporting information

**S1 Fig. Surface plots for the parameter estimates of FRECL.**
(TIF)

**S2 Fig. Mean observed adjusted Rand index; and Rand index, left and right respectively, over 50 simulations.**
(TIF)

**S3 Fig. Mean true positive and true negative clustering rates, 50 simulations for FRECL.** $n = 500, 1000$.
(TIF)

**S4 Fig. Line plots with the ARI for the simulations with one replicate against the number of clusters for FRECL and the five alternative methods for simulated data sets.** $n = 500, 1000$.
(TIF)

**S5 Fig. Line plots with the RI for the simulations with one replicate against the number of clusters for FRECL and the five alternative methods for simulated data sets.** $n = 500, 1000$.
(TIF)

**S6 Fig. Line plots with the TPR for the simulations with one replicate against the number of clusters for FRECL and the five alternative methods for simulated data sets.** $n = 500, 1000$.
(TIF)

**S7 Fig. Line plots with the TNR for the simulations with one replicate against the number of clusters for FRECL and the five alternative methods for simulated data sets.** $n = 500, 1000$.
(TIF)

**S8 Fig. Line plots with the RI, left, TPR, centre, and TNR, right for the simulations with one replicate against the number of clusters for FRECL and the five alternative methods for simulated data sets.**
(TIF)

**S9 Fig. Illustration of simulated gene expressions during 48 hours, see horizontal axis, based on [60]'s data set. Left, raw simulated values; right, smoothed values.**
(TIF)

**S1 Table. Observed sample means and standard deviations of the distributions of the % ARI in the 50 simulations.**
(TXT)

**S2 Table. Observed sample means and standard deviations of the distributions of the % ARI in the 50 simulations; comparing analyses where the alternative methods were performed with either *Y* only or *Y*, *X*. In all cases, adding the functional explanatory variables**

**results in smaller ARIs.** $K = 3$, $L_2$ distance, $n = 500$, i.i.d. random error terms.
(TXT)

**S3 Table. Observed sample means and standard deviations of the distributions of the %
ARI in the 50 simulations; comparing different numbers of individual FPCs for Alt 1 ($Y$,
$X$).** $K = 3$, $L_2$ distance, $n = 500$, i.i.d. random error terms.
(TXT)

**S4 Table. Results of the replicated simulations.** $K = 3$, $L_2$, $n = 500$, 1000, $AR(1)$.
(TXT)

**S5 Table. Observed adjusted Rand index x100 between the methods for the partitions
found setting $K = 10$ clusters in the data set with mean gene expressions spread in one day.**
The partitions from the alternative methods are calculated using $Y$, $X$.
(TXT)

**S6 Table. Observed adjusted Rand index x100 between the methods for the partitions
found setting $K = 10$ clusters in the data set with mean gene expressions spread in one day.**
The partitions from the alternative methods are calculated using $Y$.
(TXT)

**S7 Table. Observed adjusted Rand index x100 between the methods for the partitions
found setting $K = 10$ clusters in the data set with mean gene expressions in two days.** Using
all the variables $Y$, $X$ in all methods.
(TXT)

**S8 Table. Observed adjusted Rand index x100 between the methods for the partitions
found setting $K = 10$ clusters in the data set with mean gene expressions in two days.** Using
$Y$ in the alternative methods.
(TXT)

## Acknowledgments

We would like to thank Ioannis Kosmidis for helpful discussions and the Isaac Newton Institute for Mathematical Sciences, Cambridge, for support and hospitality during the programme Statistical Scalability where work on this paper was undertaken.

## Author Contributions

**Conceptualization:** Shahin Tavakoli, Daphne Ezer.

**Data curation:** Susana Conde.

**Formal analysis:** Susana Conde, Daphne Ezer.

**Funding acquisition:** Daphne Ezer.

**Investigation:** Susana Conde, Shahin Tavakoli, Daphne Ezer.

**Methodology:** Susana Conde, Shahin Tavakoli, Daphne Ezer.

**Project administration:** Shahin Tavakoli, Daphne Ezer.

**Software:** Susana Conde, Daphne Ezer.

**Supervision:** Daphne Ezer.

**Writing – original draft:** Susana Conde, Shahin Tavakoli, Daphne Ezer.

**Writing – review & editing:** Susana Conde, Shahin Tavakoli, Daphne Ezer.

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
