## [Decision Letter · Decision Letter 0]

30 May 2024

PONE-D-24-08404Functional regression clustering with multiple functional gene expressionsPLOS ONE

Dear Dr. Ezer,

Thank you for submitting your manuscript to PLOS ONE. After careful consideration, we feel that it has merit but does not fully meet PLOS ONE’s publication criteria as it currently stands. Therefore, we invite you to submit a revised version of the manuscript that addresses the points raised during the review process.

We look forward to receiving your revised manuscript.

Kind regards,

Ruofei Du, PhD

Academic Editor

PLOS ONE

Journal Requirements:

2. Please expand the acronym “EPSRC and BBSRC” (as indicated in your financial disclosure) so that it states the name of your funders in full.

3. Thank you for stating the following in the Acknowledgments Section of your manuscript: "This project was funded by the Alan Turing Institute Research Fellowship under EPSRC Research grant (TU/A/000017) to DE;  SRC/BBSRC Innovation Fellowship (EP/S001360/1) to DE and SC. ST would like o thank the Isaac Newton Institute for Mathematical Sciences, Cambridge, for support 

and hospitality during the programme Statistical Scalability where work on this paper

was undertaken. This work was supported by EPSRC grant no EP/R014604/1."

Please remove any funding-related text from the manuscript and let us know how you would like to update your Funding Statement. Currently, your Funding Statement reads as follows: "This project was funded by the Alan Turing Institute Research Fellowship under EPSRC Research grant (TU/A/000017) to DE;

EPSRC/BBSRC Innovation Fellowship (EP/S001360/1) to DE and SC. ST would like to thank the Isaac Newton Institute for Mathematical Sciences, Cambridge, for support and hospitality during the programme Statistical Scalability where work on this paper was undertaken. This work was supported by EPSRC grant no EP/R014604/1.

Engineering and Physical Sciences Research Council (EPSRC): https://www.ukri.org/councils/epsrc/

Alan Turing Institute: https://www.turing.ac.uk/

Biotechnology and Biological Sciences Research Council (BBSRC): https://www.ukri.org/councils/bbsrc/

Isaac Newton Institute for Mathematical Sciences: https://www.newton.ac.uk/

The funders did not play any role in the study design, data collection and analysis, decision to publish or preparation of the manuscript."

4. We notice that your supplementary figures are uploaded with the file type 'Figure'. Please amend the file type to 'Supporting Information'. Please ensure that each Supporting Information file has a legend listed in the manuscript after the references list.

Additional Editor Comments:

The authors please be aware of an inconsistency regarding the corresponding author's name. In the submission process, the corresponding author is identified as Daphne Ezer, but elsewhere, including in the cover letter and the manuscript, the corresponding author is identified as Susana Conde. Please help straighten this out.

I tried to access the R code mentioned in the manuscript at https://gitlab.com/sconde778/frmm_rpackagefunctions.git. However, I was prompted to create a personal account and then encountered a 404 page. If possible, please save your R code in an easily accessible web repository, such as GitHub. Ensure your code sharing is functional to facilitate the review process moving forward smoothly.

Please also be aware that one of the reviewer's comments is provided in an attachment. If you cannot see it, please contact PLOS ONE for assistance.

Reviewers' comments:

Reviewer's Responses to Questions

**Comments to the Author**

1. Is the manuscript technically sound, and do the data support the conclusions?

Reviewer #1: Yes

Reviewer #2: Yes

2. Has the statistical analysis been performed appropriately and rigorously? 

Reviewer #1: Yes

Reviewer #2: Yes

3. Have the authors made all data underlying the findings in their manuscript fully available?

Reviewer #1: Yes

Reviewer #2: Yes

4. Is the manuscript presented in an intelligible fashion and written in standard English?

Reviewer #1: Yes

Reviewer #2: Yes

5. Review Comments to the Author

Reviewer #1: Review of “Functional regression clustering with multiple functional gene expressions”

Summary

This manuscript presents a novel clustering method, functional regression clustering (FRECL), for analyzing temporal gene expression profiles. FRECL utilizes a concurrent functional linear regression model and is inspired by the traditional K-means clustering algorithm. Focusing on the associations between response and explanatory variables, the method updates each gene’s cluster label based on the best-fitted functional regression model with the smallest residual norm, aiming to identify groups of differentially expressed genes linked to biological processes over time. By treating temporal gene expression observations as functions, FRECL effectively groups genes with varying temporal expression profiles under different conditions while capturing similar changes across treatments. This approach provides a new tool for understanding gene expression dynamics and their biological implications. The authors did a good job investigating algorithm properties, i.e., the effect of the number of clusters, sample size, sensitivity study, convergence, etc. However, I hope some major and minor concerns can be addressed.

Major comments

In Line 89, the manuscript discusses the extension to settings with distinct domains for the response and each predictor. It would be better if the discussion section re-commented on this possible extension to different domains.

In Line 105, equation (4). It would be clearer if math notations were used to explain Y and functional parameters beta. Also, the Phi function used in equation (4) might be different from the equation above. A different notation for better explanations should be considered.

In Lines 263-265, L1 and L2 norms are mentioned in (iv). Although the results can be put into the supplementary material, a discussion about L1 and L2 norms is still meaningful in the main text.

Related to the above comments, in the main text simulation results display, the L2 norm was automatically chosen for every sub-section. It would be better to state why only the L2 norm is used in other simulation comparisons.

In the “Comparison with other methods” section, it will also be interesting to see the performance of other competing methods (considering observations formed with the values of both the explanatory and response variables) in the simulation study, not only based on response variable Y.

It is interesting to investigate the convergence properties of FRECL under the IID random error. A different (might be early) convergence rate might be expected.

Minor comments

Line 54, the expectation of error should have brackets.

Lines 89 and 90, “our methods” and “each predictor” should be “our method” and “each predictor”.

Line 119, you used a new notation, k ^(i). An explanation or definition is expected in lines 123 and 124 for easier understanding.

Line 208, the “5) in 20” expression for the threshold of filtering lowly expressed genes is confusing.

Line 212, “loess” is short for Locally estimated scatterplot smoothing, which should be capitalized. Also, the approach’s full name should be stated the first time it is used in the manuscript.

Figure 5 caption, please define or clarify the definition of “final ARI”.

The arrangement of Figure 7 could be improved to enhance clarity. It would be better to use A, B, C, and D letters for the top left, top right, bottom left, and bottom right to point out each sub-figure. The current layout is confusing.

Reviewer #2: The manuscript presents a significant contribution to the field of gene expression analysis through a novel clustering method. By addressing the comments and suggestions provided, the authors can enhance the clarity, readability, and overall impact of their work. The method's robustness, supported by comprehensive validation and practical application, underscores its potential for advancing biological insights in gene expression studies.

6. PLOS authors have the option to publish the peer review history of their article (what does this mean?). If published, this will include your full peer review and any attached files.

Reviewer #1: No

Reviewer #2: **Yes: **Xinmin Chu

---

## [Author Response · Author response to Decision Letter 0]

2 Aug 2024

Please view uploaded attachment for nicely formatted responses in Word, which are probably easier to read. Let us know if there are any additional issues that need addressing.

PONE-D-24-08404

Functional regression clustering with multiple functional gene expressions

PLOS ONE

Journal Requirements:

0.1 Please ensure that your manuscript meets PLOS ONE's style requirements, including those for file naming. The PLOS ONE style templates can be found at 

This has been completed

0.2 Please expand the acronym “EPSRC and BBSRC” (as indicated in your financial disclosure) so that it states the name of your funders in full.

-EPSRC and BBSRC are now named in full in the revised cover letter. Thank you for changing the submission form on our behalf. We have removed the funding information from the acknowledgement section of the manuscript as specified in 0.3.

0.3 Thank you for stating the following in the Acknowledgments Section of your manuscript: "This project was funded by the Alan Turing Institute Research Fellowship under EPSRC Research grant (TU/A/000017) to DE; SRC/BBSRC Innovation Fellowship (EP/S001360/1) to DE and SC. ST would like o thank the Isaac Newton Institute for Mathematical Sciences, Cambridge, for support 

and hospitality during the programme Statistical Scalability where work on this paper

was undertaken. This work was supported by EPSRC grant no EP/R014604/1."

Please remove any funding-related text from the manuscript and let us know how you would like to update your Funding Statement. Currently, your Funding Statement reads as follows: "This project was funded by the Alan Turing Institute Research Fellowship under EPSRC Research grant (TU/A/000017) to DE;

EPSRC/BBSRC Innovation Fellowship (EP/S001360/1) to DE and SC. ST would like to thank the Isaac Newton Institute for Mathematical Sciences, Cambridge, for support and hospitality during the programme Statistical Scalability where work on this paper was undertaken. This work was supported by EPSRC grant no EP/R014604/1.

Engineering and Physical Sciences Research Council (EPSRC): https://www.ukri.org/councils/epsrc/

Alan Turing Institute: https://www.turing.ac.uk/

Biotechnology and Biological Sciences Research Council (BBSRC): https://www.ukri.org/councils/bbsrc/

Isaac Newton Institute for Mathematical Sciences: https://www.newton.ac.uk/

The funders did not play any role in the study design, data collection and analysis, decision to publish or preparation of the manuscript."

We now include our amended funding statements in the cover letter and have removed the funding information from the acknowledgements section.

0.4 We notice that your supplementary figures are uploaded with the file type 'Figure'. Please amend the file type to 'Supporting Information'. Please ensure that each Supporting Information file has a legend listed in the manuscript after the references list.

We now upload the supplementary figures under the file type supporting information.

We ensure that each SI file has a legend listed in the manuscript after the reference list.

Additional Editor Comments:

0.5 The authors please be aware of an inconsistency regarding the corresponding author's name. In the submission process, the corresponding author is identified as Daphne Ezer, but elsewhere, including in the cover letter and the manuscript, the corresponding author is identified as Susana Conde. Please help straighten this out.

This has been straightened out. Ezer is the corresponding author and Conde is the first author. This is consistent with our roles in the project, as Conde led the implementation and testing of the solution and Ezer developed the core algorithm and led the project management. Ezer also has a more consistent e-mail address for addressing queries than Conde.

0.6 I tried to access the R code mentioned in the manuscript at https://gitlab.com/sconde778/frmm_rpackagefunctions.git. However, I was prompted to create a personal account and then encountered a 404 page. If possible, please save your R code in an easily accessible web repository, such as GitHub. Ensure your code sharing is functional to facilitate the review process moving forward smoothly.

We are sorry for the error: it appears that we forgot to change the privacy settings in the Gitlab repository. However, now we have ported the whole project to Github, so that it is part of the Ezer Lab main repository, giving it more visibility. It is now available on: https://github.com/stressedplants/FRMM

Please also be aware that one of the reviewer's comments is provided in an attachment. If you cannot see it, please contact PLOS ONE for assistance.

Reviewers' comments:

Reviewer's Responses to Questions

Comments to the Author

Reviewer #1: Review of “Functional regression clustering with multiple functional gene expressions”

Summary

This manuscript presents a novel clustering method, functional regression clustering (FRECL), for analyzing temporal gene expression profiles. FRECL utilizes a concurrent functional linear regression model and is inspired by the traditional K-means clustering algorithm. Focusing on the associations between response and explanatory variables, the method updates each gene’s cluster label based on the best-fitted functional regression model with the smallest residual norm, aiming to identify groups of differentially expressed genes linked to biological processes over time. By treating temporal gene expression observations as functions, FRECL effectively groups genes with varying temporal expression profiles under different conditions while capturing similar changes across treatments. This approach provides a new tool for understanding gene expression dynamics and their biological implications. The authors did a good job investigating algorithm properties, i.e., the effect of the number of clusters, sample size, sensitivity study, convergence, etc. However, I hope some major and minor concerns can be addressed.

Major comments

1.1 In Line 89, the manuscript discusses the extension to settings with distinct domains for the response and each predictor. It would be better if the discussion section re-commented on this possible extension to different domains.

Indeed line 89 states: ‘While we assume that the functional responses and predictors are all defined on the same domain T , our methods can easily be extended to settings with distinct domains for the response and each predictors’ We agree that it would be useful to revisit this statement in the discussion section.

We have added an additional statement on line 487: “There are several avenues by which FRECL could be extended in the future. For instance, our current implementation assumes that the response and predictors have the same domain. However, it will be possible to extend the method by first scaling the time domains, an extension that would not fundamentally alter the algorithm.”

1.2 In Line 105, equation (4). It would be clearer if math notations were used to explain Y and functional parameters beta. Also, the Phi function used in equation (4) might be different from the equation above. A different notation for better explanations should be considered.

Thank you for pointing out areas where the clarity of our mathematical explanations could improve. Please note that equation (4) is the same equation as equation (2), but written in a more compact (but standard) notation. We feel that it is useful to have both because it highlights the main structure of the model. However, we agree that the way that we worded it makes the link between Equation (2) and (4) obscure. To clarify, we have added the following text on line 107:

“Note that equation 4 is just a compact representation of equation 4.”

We also thank the reviewer for pointing out that we forgot to explicitly state that the Phi function must be the same in all these equations:

Line 107: “$\\Phi$ links the covariate $X_{il}$ and the regression slopes $\\beta$, so by using a generic $\\beta$ in (4) we allow the conditional mean to be equal to any of the conditional means in the previous equations.”

1.3 In Lines 263-265, L1 and L2 norms are mentioned in (iv). Although the results can be put into the supplementary material, a discussion about L1 and L2 norms is still meaningful in the main text.

Indeed section (iv) on line 264 states: ‘whether L1 or L2 norms are used in FRECL when computing the residuals in an iteration, see equation (5); we generate 50 simulations for each of the scenarios with K = 3; these results are included in the supplementary material.’ 

We now include a definition of L1 and L2 norms in the main text, as well as a brief discussion about choosing one or the other:

Line 269: whether the L1 or L2 norms are used in FRECL, see equation (5), to quantify the 269 magnitude of the fitted residuals in an iteration. Recall that for a function f(s), 270 s ∈ T , its L1 norm is given by R T |f(s)|ds whereas its L2 norm is [R T (f(s))2ds] 1/2 . 271 We generate 50 simulations for each of the scenarios with K = 3; these results are 272 included S2 Fig. As the L1 and L2 norms performed nearly identically, we chose 273 to only use the L2 norm in the remainder of the manuscript.

Line 490: Selection of hyperparameters, such as the choice of using $L_1$ and $L_2$ norms may depend on the specific application and will need to be re-assessed when applying the method to new data sets.

1.4 Related to the above comments, in the main text simulation results display, the L2 norm was automatically chosen for every sub-section. It would be better to state why only the L2 norm is used in other simulation comparisons.

Thank you for this suggestion:

Line 273: As both $L_1$ and $L_2$ performed nearly identically, we chose to only use $L_2$ in the remainder of the manuscript.

1.5 In the “Comparison with other methods” section, it will also be interesting to see the performance of other competing methods (considering observations formed with the values of both the explanatory and response variables) in the simulation study, not only based on response variable Y.

The reviewer is absolutely correct that both comparisons are necessary for validating the method. We actually already provided this data in S2 Table, but we forgot to reference this in the main text. Our graphs only show the best-performing versions of the algorithms (using Y only) to improve clarity. We now reference it in the text:

Line 302: When we compare methods for the simulated data, we use $Y$ only in the main text, but the methods were also evaluated with $X$ and $Y$ appended together, which is shown in S2 Table.

1.6 It is interesting to investigate the convergence properties of FRECL under the IID random error. A different (might be early) convergence rate might be expected.

FRECL performed best under AR(1) which is why we chose to focus on this scenario when we modelled convergence (see Fig 3). One of the underlying assumptions of the FRECL method is that the K-means algorithm must converge. Our aim in the convergence results section was to experimentally verify that this property was met and also to highlight to the reader that they need to experimentally test convergence in their own dataset. These conclusions are already successfully presented in the manuscript, as is. 

While we agree that there may be different convergence properties under different kinds of random error, we did not feel like this would influence the reader’s confidence in FRECL, nor influence FRECL’s usability. Therefore, we feel like this analysis– although potentially interesting– lies beyond the scope of this project. 

Minor comments

1.7 Line 54, the expectation of error should have brackets.

Good catch– fixed.

1.8 Lines 89 and 90, “our methods” and “each predictor” should be “our method” and “each predictor”.

Typos now fixed.

1.9 Line 119, you used a new notation, k ^(i). An explanation or definition is expected in lines 123 and 124 for easier understanding.This now reads: 

Line 122: Reassign the $i$th observation, $i=1,\\ldots,m$ to the best fitting model, which we refer to as $\\hat k(i)$, a function that returns the assigned cluster designation for each observation: $\\hat k(i) \\in \\{1,\\ldots, K\\}$, thus defining a new partition $P^+$.

1.10 Line 208, the “5) in 20” expression for the threshold of filtering lowly expressed genes is confusing.

This section was expanded for clarity:

Line 210: Some genes were lowly expressed in nearly all time points. To filter these out, we selected genes that were expressed at moderate levels in 20 or more time points in each season. We define moderate expression levels as those that surpass $5$ transcripts per million (TPM), which is a unit of gene expression after normalising RNA-seq data by the sequencing library depth and gene size. Using a TPM threshold of 5 is a common strategy used in biology for filtering out very lowly expressed genes \\cite{Ezer2017}

1.11 Line 212, “loess” is short for Locally estimated scatterplot smoothing, which should be capitalized. Also, the approach’s full name should be stated the first time it is used in the manuscript.

LOESS is now capitalised everywhere and the full name is included on first mention.

1.12 Figure 5 caption, please define or clarify the definition of “final ARI”.

We have added the following text to the caption: final ARI after performing consensus clustering on the individual runs.

1.13 The arrangement of Figure 7 could be improved to enhance clarity. It would be better to use A, B, C, and D letters for the top left, top right, bottom left, and bottom right to point out each sub-figure. The current layout is confusing.

This has been implemented and we agree that it looks much better. The legend now reads:

 On the basis of the elbow method, we selected K=10 clusters, both for the data set with 23 time points spread over one day (A); and for the data set with 24 time points spread over two days (B). The clusters are approximately evenly sized, under both conditions tested: over one day (C) or two days (D).

Reviewer #2: The manuscript presents a significant contribution to the field of gene expression analysis through a novel clustering method. By addressing the comments and suggestions provided, the authors can enhance the clarity, readability, and overall impact of their work. The method's robustness, supported by comprehensive validation and practical application, underscores its potential for advancing biological insights in gene expression studies.

This manuscript presents a novel method for clustering gene expression data using a function-on-function regression model. The proposed method aims to identify clusters of genes with similar changes in their expression profiles under different experimental conditions, providing useful biological insights. The manuscript includes a detailed methodology, validation through simulations, and an application to real gene expression data.

It is a well-written manuscript with several strengths. Firstly, the manuscript introduces a novel clustering method that leverages functional regression models, allowing for multiple functional explanatory variables. This approach is innovative and addresses a gap in the current methods for clustering gene expression data. Secondly, the authors validate their method through extensive simulations, demo

---

## [Decision Letter · Decision Letter 1]

27 Aug 2024

Functional regression clustering with multiple functional gene expressions

PONE-D-24-08404R1

Dear Dr. Ezer,

We’re pleased to inform you that your manuscript has been judged scientifically suitable for publication and will be formally accepted for publication once it meets all outstanding technical requirements.

Kind regards,

Ruofei Du, PhD

Academic Editor

PLOS ONE

Additional Editor Comments (optional): Please pay attention to a few of minor suggestions from the reviewers.

Reviewers' comments:

Reviewer's Responses to Questions

**Comments to the Author**

1. If the authors have adequately addressed your comments raised in a previous round of review and you feel that this manuscript is now acceptable for publication, you may indicate that here to bypass the “Comments to the Author” section, enter your conflict of interest statement in the “Confidential to Editor” section, and submit your "Accept" recommendation.

Reviewer #1: All comments have been addressed

Reviewer #2: All comments have been addressed

2. Is the manuscript technically sound, and do the data support the conclusions?

Reviewer #1: Yes

Reviewer #2: Yes

3. Has the statistical analysis been performed appropriately and rigorously? 

Reviewer #1: Yes

Reviewer #2: Yes

4. Have the authors made all data underlying the findings in their manuscript fully available?

Reviewer #1: Yes

Reviewer #2: Yes

5. Is the manuscript presented in an intelligible fashion and written in standard English?

Reviewer #1: Yes

Reviewer #2: Yes

6. Review Comments to the Author

Reviewer #1: line 115 in Revised Manuscript with Track changes: the expectation of error should have parentheses.

Reviewer #2: The revised manuscript is reasonable and clear. The authors showed details on parameter selection, discussion on computational efficiency, and expansion with recent work that highlights the significance of the biological pathways.

7. PLOS authors have the option to publish the peer review history of their article (what does this mean?). If published, this will include your full peer review and any attached files.

Reviewer #1: No

Reviewer #2: **Yes: **Xinmin Chu

---

## [Editor Report · Acceptance letter]

16 Sep 2024

PONE-D-24-08404R1 

PLOS ONE

Dear Dr. Ezer, 

I'm pleased to inform you that your manuscript has been deemed suitable for publication in PLOS ONE. Congratulations! Your manuscript is now being handed over to our production team.

Kind regards, 

on behalf of

Dr. Ruofei Du 

Academic Editor

PLOS ONE